# Codon Usage for Genetic Diversity, and Evolutionary Dynamics of Novel Porcine Parvoviruses 2 through 7 (PPV2–PPV7)

**DOI:** 10.3390/v14020170

**Published:** 2022-01-18

**Authors:** Changzhan Xie, Yimo Tao, Ying Zhang, Ping Zhang, Xiangyu Zhu, Zhuo Ha, He Zhang, Yubiao Xie, Xianzhu Xia, Ningyi Jin, Huijun Lu

**Affiliations:** 1Changchun Veterinary Research Institute, Chinese Academy of Agricultural Sciences, Changchun 130117, China; xiechangzhan2019@163.com (C.X.); tym1104597270@163.com (Y.T.); zhuxiangyu00@126.com (X.Z.); hazhuo163@163.com (Z.H.); hezhangvs@126.com (H.Z.); 877216599@163.com (Y.X.); 2College of Animal Science and Technology, Guangxi University, Nanning 530004, China; 3College of Veterinary Medicine, Jilin University, Changchun 130062, China; zhangying636@126.com; 4Institute of Specialty, Chinese Academy of Agricultural Sciences, Changchun 130117, China; zhangping@caas.cn; 5Jiangsu Co-Innovation Center for the Prevention and Control of Important Animal Infectious Disease and Zoonoses, Yangzhou University, Yangzhou 225009, China

**Keywords:** porcine parvovirus (PPV), sub-types, codon usage preferences, Bayesian method

## Abstract

Porcine parvovirus (PPV) is the main pathogen of reproductive disorders. In recent years, a new type of porcine parvovirus has been discovered and named porcine parvovirus 2 to 7 (PPV2–PPV7), and it is associated with porcine circovirus type 2 in pigs. Codon usage patterns and their effects on the evolution and host adaptation of different PPV sub-types are still largely unknown. Here, we define six main sub-types based on the Bayesian method of structural proteins of each sub-type of PPV, including PPV2, PPV3, PPV4, PPV5, PPV6, and PPV7, which show different degrees of codon usage preferences. The effective number of codons (ENC) indicates that all PPV sub-types have low codon bias. According to the codon adaptation index (CAI), PPV3 and PPV7 have the highest similarity with the host, which is related to the main popular tendency of the host in the field; according to the frequency of optimal codons (FOP), PPV7 has the highest frequency of optimal codons, indicating the most frequently used codons in its genes; and according to the relative codon deoptimization index (RCDI), PPV3 has a higher degree. Therefore, it is determined that mutational stress has a certain impact on the codon usage preference of PPV genes, and natural selection plays a very decisive and dominant role in the codon usage pattern. Our research provides a new perspective on the evolution of porcine parvovirus (PPV) and may help provide a new method for future research on the origin, evolutionary model, and host adaptation of PPV.

## 1. Introduction

The family *Parvoviridae* encompasses small non-enveloped and negative single-stranded DNA viruses, including many human and animal pathogens, that infect a wide variety of species [1]. Porcine *parvoviruses* (PPV) are important pathogens that cause reproductive failure in swine, resulting in enormous losses in the pig industry worldwide [2]. The *Parvoviridae* family consists of three subfamilies: *Parvovirinae*, Hamaparvovirinae, and *Densovirinae*. Viruses infecting vertebrates and vertebrate cell cultures are assigned to the subfamily *Parvovirinae*, and *Bocaparvovirus*, *Copiparvovirus*, and *Tetraparvovirus* contain viruses that infect pigs. Currently, eight species in four genera of parvovirus have been described to infect swine. Over the past few years, several new parvoviruses have been discovered in pigs. 

Amino acids are encoded in the form of triplet codons. An amino acid can be encoded by one or more (not more than six) triplet codons. Synonymous codons are not randomly selected within or between genomes, which is called codon bias [3,4], Codon bias is affected by many factors, including natural selection, mutation pressure, genetic drift, protein structure and characteristics, tRNA abundance, GC content, etc. [5,6,7,8,9,10]. The non-random use of synonymous codons is called “codon usage bias” [11]. It can reveal the basic characteristics of molecular evolution [12]. Additionally, the use of viral codons may be affected by the host [13,14]. Similar codon usage patterns between virus and host will severely hinder host translation. In addition, codon bias can also affect protein function and translation efficiency to affect viral protein synthesis efficiency, replication adaptability, virulence, and even virus survival [14,15,16]. Considering the dependence of the virus on host cell mechanisms, codon usage patterns can help us understand molecular evolution and expression levels and help vaccine design [17]. Recombination analysis of other pathogens with other DNA [18] and RNA [19] viruses, including recombination outages and evolutionary rates. 

Until now, the influence of codon usage bias on the genetic evolution of PPV has not been elucidated in detail. We conducted a comprehensive analysis of the evolutionary process, genetic evolution, and codon usage patterns based on the updated global-scale PPV sequence data set. Our research not only provides detailed information about the codon usage in PPV (PPV2-NS1, PPV3-VP1, PPV4-CP, PPV5-NS1, PPV6-NS1, and PPV7-NS1) genes, but it also demonstrates PPV evolution, species spread, and host adaptation to provide a novel approach.

## 2. Materials and Methods

### 2.1. Genome Alignment and Phylogenetic Analysis

The nucleotide and deduced amino acid (AA) sequences of PPVs were assembled using the SeqMan program of DNAstar software version 7.0 (DNASTAR Inc., Madison, WI, USA) to determine sequence homology and genetic variations. The phylogenetic analysis is based on the nucleotide sequence by the distance-based neighbor-joining method using the iTOL v6 with bootstrap analysis of 1000 replicates (https://itol.embl.de/, accessed on 26 November 2021). All reference sequence information is listed in Appendix A (the nucleotide database is recorded as of 2021).

### 2.2. Phylogeographic Model

jModelTest software selected the best evolution model, and BEAST v1.10.4 was used under the GTR + I + G model of nucleotide substitution with a relaxed clock. The PPV gene was used for phylogenetic and phylodynamic analysis of PPV, and its high variability compared with other genes and having the most available sequences. Markov chain Monte Carlo (MCMC) (set to perform 80 million computations) methods were used to analyze a maximum clade credibility tree. The BEAUti software (version 1.10.4) (http://beast.community/beauti#bayesian-evolutionary-analysis-utility-beauti) (accessed on 1 November 2021) is used to estimate the time to the nearest common ancestor (tMRCA) and the rate of evolution [20]. The best-fit model (GTR + I + G), a relaxed clock (log-normal), and a coalescent Bayesian skyline model was set to estimate the efficient population size. The length of the chain was 1 × 10^8^ generations, every echo state to screen was 1 × 10^6^, and every log parameter was 1 × 10^4^. The independent run was applied using BEAST v1.10.4 software. The analysis of the sampled data was output by Tracer v1.7 (version 1.7.2) (http://beast.community/tracer, accessed on 16 November 2021), and the results of the MCMC tree model were output by the Tree Annotator program.

### 2.3. Codon Usage Index

#### 2.3.1. Nucleotide Composition

The nucleotide composition of the PPVs (PPV2-NS1, PPV3-VP1, PPV4-CP, PPV5-NS1, PPV6-NS1, and PPV7-NS1), including the content of each nucleotide (A%, T%, C%, and G%), total GC, and the AT content of each sequence, is calculated using Bioedit (version 7.0.9.1). The frequency of the third nucleotide (A3s, C3s, T3s, and G3s) was counted using Codon W (version 1.4.2) (http://codonw.sourceforge.net/, accessed on 10 November 2021). GC content (%G+C) refers to the GC content at the first (GC1s), second (GC2s), and third (GC3s) codon positions (http://emboss.toulouse.inra.fr/cgi-bin/emboss/cusp, accessed on 17 November 2021). The average frequency of GC1s and GC2s (GC12s) is calculated through the use of CAIcal [21]. Stop codons (TAA, TAG, and TGA) that do not encode any amino acids and non-degenerate ATG and TGG codons were excluded from the analysis.

#### 2.3.2. Effective Number of Codons (ENC) Analysis

The effective number of codons (ENC) describes the degree of deviation of codon usage from random selection. It is not a comparison of the frequency of a particular codon with other codons, but it reflects the unbalanced use of synonymous codons in the codon family degree of preference. Calculated based only on codon usage data, it has nothing to do with gene length and amino acid (aa) composition [22] (detailed introduction in Appendix A).

#### 2.3.3. Relative Synonymous Codon Usage Analysis (RSCU)

Relative synonymous codon usage (RSCU) is used to check the usage of synonymous codons to assess the deviation of synonymous codon usage and to highlight the relative frequency of synonymous codons without being affected by the confounding of the amino acid composition of different gene products [23]. In research, RSCU values of highly expressed genes are usually used to establish a reference table (http://www.bioinformatics.nl/cgi-bin/emboss/cusp, accessed on 2 November 2021) (detailed introduction in Appendix A).

#### 2.3.4. Relative Codon Deoptimization Index (RCDI)

The relative codon deoptimization index (RCDI) is a measure of comparison with the general codon distribution. The translation rate of viral genes can be estimated, especially the translation rate of the entire genome; the higher the similarity between the virus and the host gene (close to the RCDI value of 1), the higher the translation rate [24]. The RCDI value is calculated by CAIcal (http://genomes.urv.cat/CAIcal/RCDI, accessed on 8 November 2021) [21]. In addition, high RCDI may also indicate that certain genes are expressed during the incubation period, and the virus may even exhibit a low replication rate (detailed introduction in Appendix A).

#### 2.3.5. Codon Adaptation Index (CAI) Analysis

The CAI value ranges from 0 to 1. The higher the CAI value, the higher the similarity to the host [25] (detailed introduction in Appendix A).

#### 2.3.6. Frequency of Optimal Codons (FOP) Analysis

The optimal codon for an amino acid is defined as the codon with the largest number of tRNA genes with its anticodon [23]. Therefore, the FOP metric is a weighted average of the best codon RSCU, where the weight is the corresponding amino acid frequency (detailed introduction in Appendix A).

#### 2.3.7. Codon Bias Index (CBI) Analysis

Codon bias index (CBI) reflects the composition of high-expression superior codons in a specific gene. For the target host’s own genes, this index has a good correlation with the ENC value, which more clearly reflects the expression of foreign genes in the target host. The value of CBI ranges from 0 to 1. When translating amino acids, the number of synonymous codons is the same, and the CBI is 1. When an amino acid only has one corresponding codon, that is, the codon preference is maximized, then the CBI is 0. It is also possible that the number of occurrences of the optimal codon is less than the average number of usages, the CBI is a negative value [26] (detailed introduction in Appendix A).

#### 2.3.8. ENC–GC3s Drawing Analysis

By exploring the functional relationship between ENC (ordinate) and GC3s (abscissa), it is tested whether there are other factors involved in the formation of codon usage patterns in addition to mutation pressure. If there is no natural selection, codon preference is only subject to mutation pressure. In an ideal state, the point corresponding to the gene will fall on the desired curve or a position closer to the curve. When the formation plays an important role, the ENC value is located or distributed around the expected curve; however, when the use of codons is affected by natural selection, mutation pressure, and other factors, the ENC value is much lower than the expected curve [27], indicating that other factors are also involved in codon use preference.

#### 2.3.9. Neutrality Plot Analysis (G12s/GC3s) 

The linear relationship between GC12 and GC3 by neutrality graph analysis, which is to study the relative contribution of mutation and natural selection pressure to codon usage bias. If the slope of the equation is close to 1, it means that sudden change pressure is the dominant factor and has a strong dependence [10]. Conversely, values close to the abscissa or ordinate indicate that natural selection is dominant.

### 2.4. Statistical Analysis 

The scores of CAI, RSCU, RCDI, FOP, and CBI are not strictly normally distributed; the pedigrees have unequal variances, and their scores have statistically significant differences. Significant relationships are shown in box plots, where *p* ≤ 0.001 is extremely significant (****).

## 3. Results 

### 3.1. Phylogenetic Analysis of PPV_S_

To establish the genetic relationships of strains of PPV, a phylogenetic analysis of PPVs was performed based on the nucleotide sequences of the structural protein using a distance-based neighbor-joining method with 1000 bootstrap replicates in iTOL v6 (Figure 1). 

### 3.2. Evolutionary Tree Construction Based on the Bayesian Markov Chain Method

The Bayesian Markov chain method was used to determine the mutation rate of the PPV strain’s structural protein gene codons, and the BEAST results were analyzed by Trace. The results showed that the mutation rates of the amino acid codons encoded by the PPV strain’s structural protein gene were different, and the mutation rates of the three codons were 1.0214, 0.8584, and 1.1203 (Figure 2A). Therefore, the third codon has the highest mutation rate. Because the third codon has the highest mutation rate and the codon has degeneracy, some mutations will not change the amino acid encoding the protein, which makes the homology between PPV virus strains very high. These values indicate that during this period, the structural protein gene has a base mutation, which may be related to the recent high detection rate of the PPV virus. The geographical distribution of the PPV virus is steadily increasing. In the skyline chart, it can be seen that the effective outbreak of PPV virus has declined since 2007, but it also increased in 2015 (Figure 2B,C).

### 3.3. Nucleotide Bias of PPV Genotypes

Among PPV antigenic variants, Table 1 shows that nucleotide A (0.272 ± 0.005 in all) is the most frequent of all the strains, with PPV2, PPV3, PPV4, PPV5, PPV6, and PPV7 showing percentages of 0.231 ± 0.002, 0.231 ± 0.002, 0.342 ± 0.001, 0.345 ± 0.002, 0.235 ± 0.001, and 0.265 ± 0.005 (mean ± SD), respectively. in terms of synonymous codons, while the mean value of T3s (0.347 ± 0.152 in all) is the highest at the third codon position, followed by A3s (0.329 ± 0.129 in all), C3s (0.299 ± 0.123 in all), and G3s (0.284 ± 0.078 in all), the third position of synonymous codons shows dissimilar composition patterns among these PPV sub-types. Furthermore, the GC content varied among different synonymous codon positions and was also different among PPV sub-types. While the value of GC3_S_ in PPV2 and PPV7 is higher than GC1_S_ and GC2_S_, the GC3_S_ of other sub-types is lower than GC1_S_. In particular, regardless of the position of the synonymous codon, PPV5 showed the lowest GC content. PPV7 has the highest GC12s values relative to other sub-types PPV2, PPV3, PPV4, PPV5, and PPV6.

### 3.4. Codon Bias Measures

The ENC values were 51.607 ± 0.891, 57.433 ± 0.192, 44.843 ± 0.270, 5.469 ± 1.007, 48.045 ± 0.325, and 45.812 ± 0.812 for PPV2, PPV3, PPV4, PPV5, PPV6, and PPV7, respectively. Although all ENC values are higher than 35, indicating that the codon biases of all sub-types are low, the ENC values of PPV2 and PPV3 can reach 51.607 (SD ± 0.891) and 57.433 (SD ± 0.192), the codons of PPV2 and PPV3 are lower and the endogenous genes corresponding to various sub-types of PPV tend to be less expressed (Table 1).

We compared the RSCU values among different PPV sub-types. RSCU analysis shows that some codons may be regarded as overrepresented (RSCU > 1.6) or be said to be underrepresented (RSCU < 0.6) in different PPV sub-types (Appendix A). Codons AGA(Arg) (2.279 ± 1.312 in all) is overrepresented in all six sub-types, and GTC(Val) (0.564 ± 0.374 in all), TCG(Ser) (0.395 ± 0.236 in all), GCG(Ala) (0.553 ± 0.211 in all), CGT(Arg) (0.439 ± 0.414 in all), and CGA(Arg) (0.528 ± 0.225 in all) are overrepresented. More specifically, codon AGA(Arg) is most overrepresented in PPV4, PPV5, and PPV7, while it is low in PPV2, PPV3, and PPV6, and codon TCG(Ser) is underrepresented in PPV3, PPV4, PPV5, PPV6, and PPV7, while it is low in PPV2; codon GCG(Ala) is underrepresented in PPV2, PPV4, PPV5, and PPV7, while it is low in PPV3 and PPV6; codon CGT(Arg) is underrepresented in PPV2, PPV3, PPV4, PPV5, and PPV7, while it is low in PPV6; codon CGA(Arg) is underrepresented in PPV2, PPV3, PPV4, PPV5, and PPV6, while it is low in PPV7. Surprisingly, the AGA(Arg) was 4.097 ± 0.129 and 3.812 ± 0.107 in PPV4 and PPV5, respectively, indicating an extremely preferential use of this codon (Appendix A).

### 3.5. Codon Usage Comparison between Virus and Host

CAI is a measure of the codon usage adaptability of the most commonly used synonymous codons in the reference genome, and it is usually used to predict gene expression efficiency and the expression level of foreign genes. Regarding the CAI value, PPV3 and PPV7 have the highest CAI values, which proves that their codon bias is very strong relative to other sub-types (Figure 3A). The relative codon deoptimization index (RCDI) is a measure of comparison with the general codon distribution. The translation rate of viral genes can be estimated, especially the translation rate of the entire genome. Among them, PPV4 has the highest RCDI value, indicating that the higher the similarity with the host gene, the higher the translation rate (Figure 3B). The optimal codon refers to the most frequently used codon in the highly expressed genes of a species. The results show that PPV7 has the highest FOP value compared to other sub-types, indicating that it has optimal codon usage frequency (Figure 3C). PPV7 was a new PPV sub-type in recent years. The CAI, RCDI, and FOP values show that it has strong expression of foreign genes and optimal codon usage compared to other PPV sub-types.

### 3.6. CBI–ENC Correlation Analysis

The CBI value reflects the composition of a gene with high expression of superior codons, but it has a good correlation with the ENC value. The CBI–ENC correlation analysis shows that the CBI value is indeed correlated with the ENC value, but only PPV2 and PPV7 have a positive CBI value, and PPV7 is the highest, indicating that the synonymous codons occur the same number of times when translating amino acids; the CBI value of other PPV3, PPV4, PPV5, and PPV6 sub-types are negative, indicating that the number of occurrences of optimal codons is lower than the average number of usages (Appendix A).

### 3.7. ENC–GC3s Drawing Analysis 

Use the ENC–GC3s diagram to explore the influence of GC3s on codon preference. As shown in Figure 4, the distributions of the ENC–GC3s diagrams of the seven PPV sub-types are relatively similar, and all genes fall under the expected curve or just on the expected curve. This means that indicating that mutation pressure is less suitable for codon usage patterns, but other factors, especially natural selection, play an important role in codon usage preference.

### 3.8. Neutrality Plot Analysis

Neutral analysis based on GC12 and GC3 can quantitatively evaluate the impact of pressure mutations and natural selection. We found that the coefficients between GC12s and GC3s are positively correlated. This phenomenon suggests that the role of natural selection may be important in shaping PPV codon usage, especially for PPV3 (slope of regression line 0.0976, R^2^ = 0.0634), PPV5 (slope of regression line 0.1518, R^2^ = 0.178), PPV6 (slope of regression line 0.1365, R^2^ = 0.167), and PPV7 (slope of regression line 0.0633, R^2^ = 0.0993) (Figure 5). This indicates that mutational pressure has a certain impact on the codon usage preference of PPV genes, and natural selection plays a very important or even leading role in the codon usage pattern.

## 4. Discussion

Several studies to determine the effects of PPV2–PPV7 on pig health have focused on concurrent infections between PPV and other pathogens (mainly PRRSV and PCV2). Because PPV2–PPV6 have never been cultured and challenged in vitro, their importance must be inferred or speculated based on testing at the DNA level. Previous phylogenetic studies of strains on a global scale have shown that they can be divided into seven different sub-types. Therefore, studying the transmission mode and gene codon bias of different sub-types of PPV will provide more objective data and help to better understand the ecological and genetic evolutionary dynamics of these viruses. In order to explore the evolutionary characteristics and host adaptability of different PPV sub-types, a codon usage analysis was performed here. Based on the phylogenetic analysis of MCC, we identified six main PPV sub-types (PPV2, PPV3, PPV4, PPV5, PPV6, and PPV7) and determined the evolutionary dynamics of PPV for the first time. 

Bayesian system dynamics models are used in research environments and are rarely used for routine monitoring of field data to support disease prevention and control. Due to the complexity and scale of the data analyzed, this implementation is challenging. Bayesian inference (BI) is a statistical method based on using evolutionary models of sequence evolution to reconstruct a phylogenetic tree. The resulting tree not only reflects the best estimate of the phylogenetic relationship but also provides definite support for branching [28]. Many infectious viral diseases use Bayesian system dynamics models. For example, the study simulated the evolutionary dynamics of PRRSV [29,30]. The data we selected include all the published structural protein sequences of various sub-types of PPV by NCBI. The skyline chart shows that the effective outbreak of PPV virus has declined since 2007 and increased in 2015. The chart can also predict the trend of dynamic phylogeny, indicating the trend of PPV outbreak, and provide a theoretical basis for clinical prevention and control.

Regarding the codon usage pattern of the PRRSV ORF5 gene, the changes in different lineages of PRRSV reflect the evolutionary changes made for survival and adaptation to the host, confirming that the attenuation of continuous passage of PRRSV in cells has a certain relationship with codon preference, so it indicates that codon preference has a certain impact on the survival, host adaptation, and virulence of PRRSV virus [31]. Previous studies have confirmed that the codon usage preference of the PCV2 clade is mainly affected by natural selection, and its evolutionary model seems to be spread across hosts [32]. It was then determined that H7N9 has a low codon bias, which is mainly driven by natural selection [33]. It proved that codons are very important for adapting to the host. In this study, based on a comprehensive analysis of CAI, FOP, and RCID, the new PPV7 sub-type has the highest CAI and FOP values compared to other sub-types, and the RCID value of PPV7 is not the highest, but it remains above the average level. Because previous studies have shown that the CAI value is high, indicating that the virus has strong adaptability [34], and inferred that it may be related to the virus’s virulence to the host [33], it has important guiding significance for studying whether the new PPV7 sub-type has higher infectivity and pathogenicity. Using the ENC–GC3s diagram to explore the codon preference of GC3s shows that the distribution of the ENC–GC3s diagrams of the seven PPV sub-types is relatively similar, indicating that mutation pressure is less suitable for codon usage patterns. The CBI-ENC correlation analysis shows that the CBI value is indeed related to the ENC value, but only the CBI values of PPV2 and PPV7 are positive, and PPV7 is the highest, indicating that the number of synonymous codons are the same with translating amino acid groups; the CBI values of other sub-types PPV3, PPV4, PPV5, and PPV6 sub-types are negative, indicating that the number of occurrences of the best codons is lower than the average number of usages. Neutral analysis based on GC12 and GC3 can quantitatively evaluate the impact of stress mutations and natural selection. We have determined that mutation stress has a certain impact on the codon usage preference of PPV genes, and natural selection plays a very important and leading role in codon usage patterns. To sum up, PPV is still widely distributed in China, and the new type of PPV7 is determined by the analysis of the bias in cryptography to be able to better adapt to the host and thus better spread.

## 5. Conclusions

In conclusion, we have determined that the codon usage preferences of PPV sub-types were mainly affected by natural selection and played an important role in the evolution process. Importantly, the main evolutionary pattern of PPV that appears to be fixed through host transmission. Our research is an effective model for the evolution, origin, and codon bias of various viruses, as well as predicting the trend of dynamic phylogeny, providing a theoretical basis for prevention in clinical practice.

## Figures and Tables

**Figure 1 viruses-14-00170-f001:**
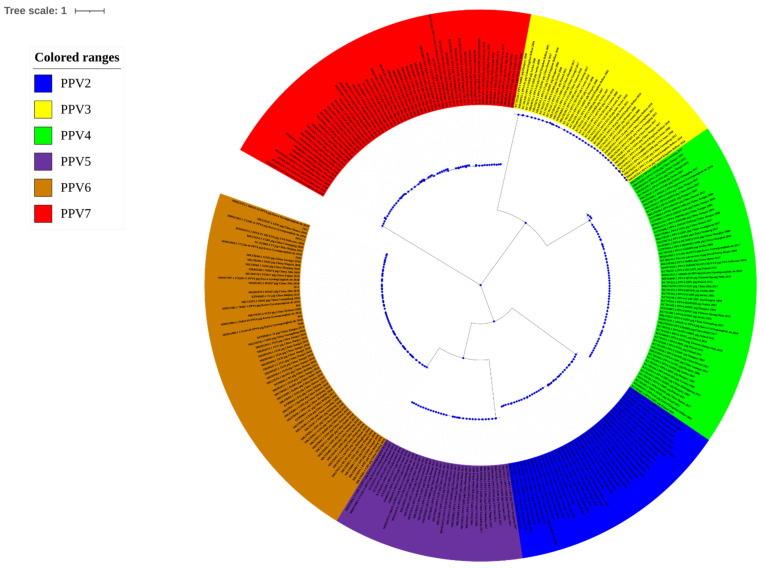
PPV phylogenetic tree based on the nucleotide sequence of all PPV sub-types.

**Figure 2 viruses-14-00170-f002:**
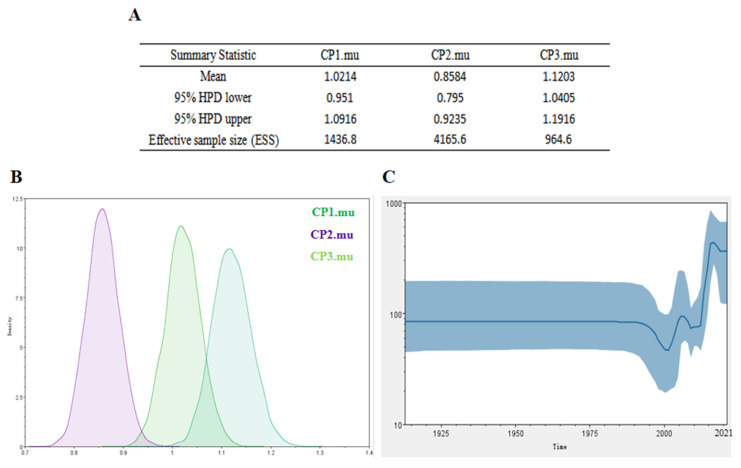
PPV virus structural protein codon mutation rate and skyline diagram (**A**,**B**). The codon mutation rate of PPV virus structural protein gene was estimated by Bayesian Markov chain method. The codon mutation rate is the result of BEAST running using Trace analysis. (**C**) The dynamic study of the genetic diversity of PPV virus structural protein genes by Bayesian skyline diagram. The thick solid line is the median estimate, and the dashed line represents the 95% confidence interval. The abscissa is time, and the ordinate is the effective population size.

**Figure 3 viruses-14-00170-f003:**
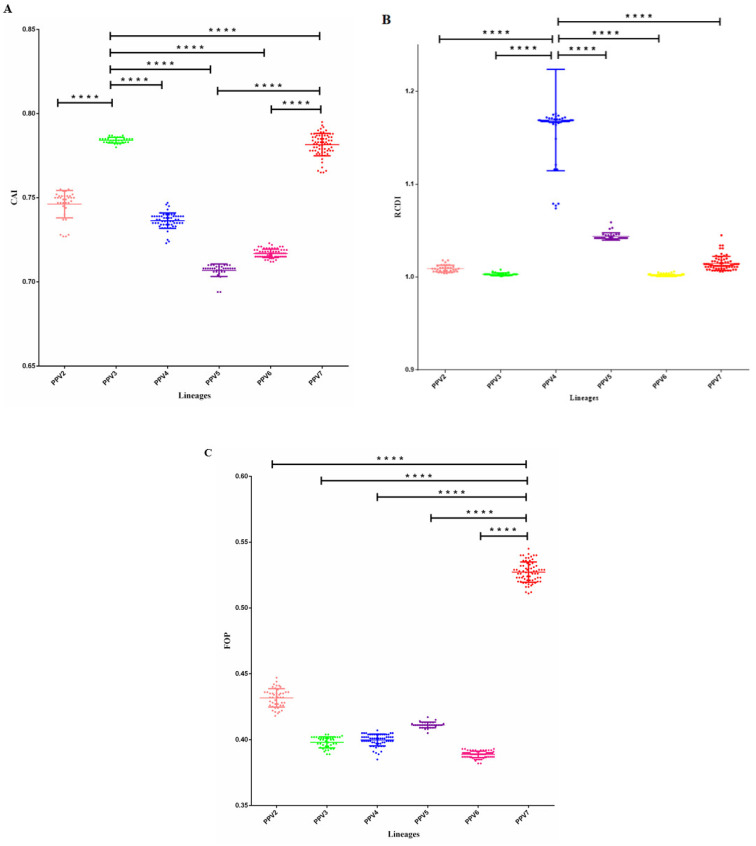
(**A**): CAI scatter plots of structural protein gene from different sub-types of all PPV strains. The asterisk indicates a significant difference between labeled groups. (**B**): RCDI scatter plots of structural protein gene from different sub-types of all PPV strains. The asterisk indicates a significant difference between labeled groups. (**C**): FOP scatter plots of structural protein gene from different sub-types of all PPV strains. The asterisk indicates a significant difference between labeled groups (*p* < 0.001 ****).

**Figure 4 viruses-14-00170-f004:**
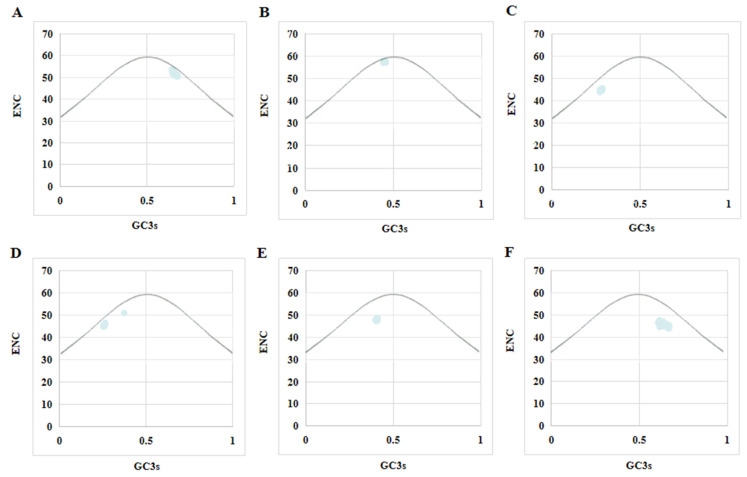
The relationship between ENC and GC3 of each sub-type of PPV. (**A**): PPV2. (**B**): PPV3. (**C**): PPV4. (**D**): PPV5. (**E**): PPV6. (**F**): PPV7.

**Figure 5 viruses-14-00170-f005:**
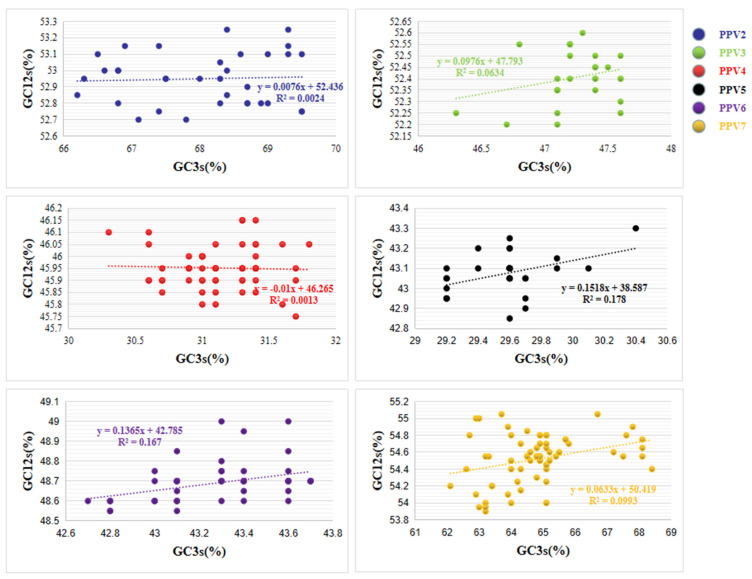
Neutrality plot of each sub-type of PPV.

**Table 1 viruses-14-00170-t001:** Properties of structural protein genes from PPV strains analyzed in this study (mean value ± SD).

Categories	PPV2	PPV3	PPV4	PPV5	PPV6	PPV7	ALL
%A	0.231 ± 0.002	0.231 ± 0.002	0.342 ± 0.001	0.345 ± 0.002	0.235 ± 0.001	0.265 ± 0.005	0.272 ± 0.005
%C	0.282 ± 0.002	0.282 ± 0.002	0.201 ± 0.001	0.170 ± 0.001	0.210 ± 0.001	0.274 ± 0.002	0.239 ± 0.004
%T	0.189 ± 0.002	0.189 ± 0.002	0.248 ± 0.001	0.270 ± 0.001	0.260 ± 0.001	0.155 ± 0.002	0.224 ± 0.005
%G	0.298 ± 0.003	0.298 ± 0.002	0.202 ± 0.001	0.212 ± 0.001	0.295 ± 0.004	0.305 ± 0.006	0.264 ± 0.004
A3_S_	0.212 ± 0.009	0.247 ± 0.004	0.506 ± 0.004	0.522 ± 0.017	0.199 ± 0.003	0.333 ± 0.017	0.329 ± 0.129
C3_S_	0.440 ± 0.008	0.266 ± 0.003	0.179 ± 0.002	0.194 ± 0.015	0.213 ± 0.004	0.473 ± 0.009	0.299 ± 0.123
T3_S_	0.202 ± 0.007	0.411 ± 0.003	0.396 ± 0.003	0.434 ± 0.010	0.532 ± 0.004	0.128 ± 0.004	0.347 ± 0.152
G3_S_	0.380 ± 0.008	0.303 ± 0.003	0.183 ± 0.003	0.150 ± 0.010	0.315 ± 0.004	0.332 ± 0.023	0.284 ± 0.078
%G+C	0.580 ± 0.003	0.507 ± 0.001	0.410 ± 0.001	0.387 ± 0.003	0.469 ± 0.001	0.580 ± 0.006	0.494 ± 0.074
GC1_S_	0.587 ± 0.003	0.593 ± 0.001	0.479 ± 0.001	0.483 ± 0.002	0.540 ± 0.001	0.621 ± 0.006	0.552 ± 0.060
GC2_S_	0.473 ± 0.002	0.455 ± 0.002	0.440 ± 0.002	0.379 ± 0.002	0.434 ± 0.001	0.470 ± 0.003	0.444 ± 0.030
GC3_S_	0.681 ± 0.010	0.472 ± 0.003	0.311 ± 0.003	0.296 ± 0.003	0.433 ± 0.003	0.648 ± 0.014	0.478 ± 0.150
GC12_S_	0.530 ± 0.002	0.524 ± 0.001	0.460 ± 0.001	0.431 ± 0.001	0.487 ± 0.001	0.545 ± 0.003	0.500 ± 0.040
ENC	51.607 ± 0.891	57.433 ± 0.192	44.843 ± 0.270	45.469 ± 1.007	48.045 ± 0.325	45.812 ± 0.812	48.319 ± 4.073
CBI	0.036 ± 0.012	−0.027 ± 0.007	−0.035 ± 0.008	−0.024 ± 0.005	−0.053 ± 0.004	0.013 ± 0.173	−0.015 ± 0.068

## Data Availability

The data presented in this study are available in the article and in its online Appendix A.

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
