# Peer review of "Codon Usage for Genetic Diversity, and Evolutionary Dynamics of Novel Porcine Parvoviruses 2 through 7 (PPV2–PPV7)"

_viruses, 2022, doi:10.3390/v14020170_

Round 1

Reviewer 1 Report

Manuscript ID: viruses-1534697

Codon usage for Genetic Diversity, and Evolutionary Dynamics of Novel Porcine Parvoviruses 2 through 7 (PPV2-PPV7)

In this study, the authors provide information about the codon usage to evaluated genetic diversity, and evolutionary dynamics from novel porcine parvoviruses (PPV2- PPV7). It is the opinion of this reviewer that the manuscript is suitable for publication on Viruses; however, minor revisions are required.

INTRODUCTION

Page 1, Lines 38-39: “The family Parvoviridae consists of two subfamilies: Parvovirinae and Densovirinae.”. However, the classification of the family Parvoviridae is out of date. I suggest the authors consult Virus Taxonomy: 2020 Release of the International Committee on Taxonomy of Viruses (ICTV) and review this information.

Author Response

Reviewer: 1

Codon usage for Genetic Diversity, and Evolutionary Dynamics of Novel Porcine Parvoviruses 2 through 7 (PPV2-PPV7)

In this study, the authors provide information about the codon usage to evaluated genetic diversity, and evolutionary dynamics from novel porcine parvoviruses (PPV2- PPV7). It is the opinion of this reviewer that the manuscript is suitable for publication on Viruses; however, minor revisions are required.

INTRODUCTION

Page 1, Lines 38-39: “The family Parvoviridae consists of two subfamilies: Parvovirinae and Densovirinae.”. However, the classification of the family Parvoviridae is out of date. I suggest the authors consult Virus Taxonomy: 2020 Release of the International Committee on Taxonomy of Viruses (ICTV) and review this information.

It is very grateful for your remind. I've re-looked up on ICTV and edited them. Please see line 52-53.

Reviewer 2 Report

This is a very interesting and important paper. There are only a few issues that need to be considered and modified.

The author should consider the recombination event among the PPV dateset. So you need to write whether you have detect the recombination in your dateset. The whole paper needs to be analyzed with un-recombination data sets.

In introduction part ,the authors need to add more describe about Recombination; evolution of DNA and RNA viruse;and Porcine Parvoviruses. Such as compare the recombination ,evolutionary rate to other DNA virus and RNA viruses (eg,coronavirus,  https://doi.org/10.1093/molbev/msab364 and https://doi.org/10.1093/molbev/msaa117 ).

Author Response

Reviewer: 2
This is a very interesting and important paper. There are only a few issues that need to be considered and modified.

The author should consider the recombination event among the PPV dateset. So you need to write whether you have detect the recombination in your dateset. The whole paper needs to be analyzed with un-recombination data sets.

In introduction part ,the authors need to add more describe about Recombination; evolution of DNA and RNA viruse;and Porcine Parvoviruses. Such as compare the recombination ,evolutionary rate to other DNA virus and RNA viruses (eg,coronavirus,  https://doi.org/10.1093/molbev/msab364 and https://doi.org/10.1093/molbev/msaa117 ).

It is very grateful for your remind. I have added the corresponding requirements. Please see line 72-74.

Reviewer 3 Report

The manuscript “Codon usage for Genetic Diversity, and Evolutionary Dynamics of Novel Porcine Parvoviruses 2 through 7 (PPV2-PPV7)” by Xie et al. describe the detailed analysis of codon usage of different subtypes of PPV and their evelutionary dynamics. Xie. et al. used different bioinformatics packages/softwares to conclude that the codon usage preference by natural selection played very important role on evolutionary process of PPV. 

This study presented by Xie et al. covered all the relevant parameters to conclude the study which make this manuscript in interest of the readers.

Author Response

The manuscript “Codon usage for Genetic Diversity, and Evolutionary Dynamics of Novel Porcine Parvoviruses 2 through 7 (PPV2-PPV7)” by Xie et al. describe the detailed analysis of codon usage of different subtypes of PPV and their evelutionary dynamics. Xie. et al. used different bioinformatics packages/softwares to conclude that the codon usage preference by natural selection played very important role on evolutionary process of PPV. 

This study presented by Xie et al. covered all the relevant parameters to conclude the study which make this manuscript in interest of the readers.

It is very grateful for your remind.